# Random Projection Variational Auto-Encoders

## Abstract

Variational Auto-Encoders optimise the parameters of a distribution that approximates the posterior distribution of some data. We focus on the case where the approximating distribution includes Gaussian distributions related to each datum. When these Gaussians are each defined on a high-dimensional space, it is often assumed that using full-rank covariance matrices would be prohibitively computationally expensive and would be prone to overfitting. In such settings, a parameterisation that constrains each covariance matrix to be diagonal is often adopted. We propose the use of approximations that offer the potential for alternative compromises between the computational expense, overfitting and accuracy of full-rank and diagonal covariances. More specifically, we propose using covariance matrices that involve a random projection of a full-rank covariance in a low-dimensional space. In this ablation study, we isolate the varying parameterisation from other techniques and assess the impact of the dimensionality of this low-dimensional space on both computational cost and accuracy in the context of MNIST, CIFAR-10 and Flowers-102. We observe that, for a finite number of training iterations, accuracy is maximised by the compromise that is neither equivalent to the full-rank covariance nor a diagonal covariance. We also identify that the computational cost fluctuates less than one might anticipate and that performance is improved with the parameterisation considering a random projection from a lower full-rank covariance.

## 1 Introduction

Machine Learning is increasingly pervasive (Ling et al., 2022; Rohrhofer et al., 2023). While it is common to optimise the parameters of the models, there is also significant interest in Bayesian Neural Networks. See for example: Antoran et al. (2023); Grinwald et al. (2023); McDermott & Wikle (2019); Gal & Ghahramani (2015). In the authors' view, this interest is well justified for two reasons. First, Bayesian Neural Networks have an inherent ability to convey confidence (and so facilitate sensitivity analysis, for example). Second, Bayesian Neural Networks have the potential to outperform alternative methods, both because they output a parameter estimate that is a mean, not a maximum, but also because they make it possible for parametric uncertainty to be considered when each test datum is processed: the output for the test datum can involve an average over the values for the parameter of the model, enabling the output to be more robust to uncertainty associated with any parameter estimate.

There are two broad families of Bayesian Neural Networks, those that use Numerical Bayesian techniques (e.g. Markov Chain Monte Carlo (Neal, 1992)) and those that use Variational approximations (MacKay, 1995). While Variational approximations necessarily restrict the uncertainty that can be expressed, they are computationally cheaper than numerical Bayesian approaches. We focus here on Variational approximations.

When using a Variational approximation, we find the best-fitting parametric model to a statistical distribution of interest (which is typically a likelihood) (Blei et al., 2017). The choice of parameterisation is crucial; having more parameters typically enables the model to be more expressive, but also increases the computational expense and the potential for overfitting. The need to balance these considerations makes it desirable to have a continuum of parameterisations and to avoid discontinuities in the quantity of parameters being considered.

The parametric models considered in the Variational approximations used in popular algorithms often assume Gaussian distributions. These Gaussians' covariance matrices can be approximated as diagonal. With $J$ dimensions, a full covariance matrix has $\frac{1}{2}J \times (J+1)$ unique parameters, whereas a diagonal covariance matrix has $J$. Only having these two parameterisations introduces a discontinuity in terms of the quantity of parameters considered. It would be preferable to have a continuum between these two parameterisations. A 'middle ground' between the theoretical robustness offered from a full rank covariance and the reduced computational complexity of the diagonal approximation.

In this paper, we focus on the parameterisation of the covariance matrices used in Variational Auto-Encoders (Kingma & Welling, 2013) (VAEs): a VAE offers a nonlinear alternative to the functionality provided by Principal Component Analysis (Jolliffe, 2002) and Linear Discriminant Analysis (Izenman, 2013). We consider VAEs a useful and widely applicable example application where the parameterisation of covariance matrices is relevant.

## 1.1 Contributions

We propose a parameterisation that aims to offer the continuum described above in the context of covariance matrices. More specifically, we propose a parameterisation of covariance matrices that is inspired by work on Random Projection (RP) approximations (Vempala, 2005). While we believe the approach is more widely applicable, we focus on the application in the context of Variational Auto-Encoders (Kingma & Welling, 2013): while Kingma & Welling (2013) makes clear that its consideration of a diagonal covariance matrix is an example of the application of the paper's methods, the paper only considers this single parameterisation in their experiments. We propose a more flexible parameterisation that we perceive to be novel in this context. We then analyse the performance (in terms of both loss and computational cost) that results from the use of this parameterisation in the context of two popular datasets, MNIST and CIFAR-10 and the more computationally demanding Flowers-102.

**The paper is structured as follows:** First, we briefly discuss how RP can be integrated within VAEs in section 3. Then, we present results on the MNIST dataset, CIFAR-10 and Flowers-102, with a focus on the information extraction of RP in section 4. We then discuss the results in section 5 before drawing conclusions in section 6. Finally, we identify opportunities for future work in section 7.

## 2 Related work

There are, of course, many papers in many application domains that consider the compromises offered by different parameterisations in terms of their accuracy and computational expense. This choice of parameterisation is closely related to the general problem of identifying compact feature representations, i.e., dimensionality reduction (Sorzano et al., 2014). Dimensionality reduction has been shown to reduce computational requirements both without negatively impacting accuracy (for example, del Águila et al. (2019) considers dimensionality reduction in a geophysics application and reports a 20-fold speed-up with no reduction in accuracy) but also in such a way that accuracy is improved (for example, Sulayes (2017) and Ren et al. (2018) report improvements in accuracy in the contexts of authorship attribution and analysis of exoplanets respectively).

As explained in Nabil (2017); Dasgupta (2013), Random Projection has also been applied in diverse application domains. While there are examples of Random Projection being used in the context of Neural Networks (see, for example, Wójcik & Kurdziel (2019); Vinh et al. (2016) and Peng et al. (2021)), these pre-existing approaches consider random projections of the data or manipulating the weights of a Neural Network. In contrast, the approach considered herein considers (somewhat implicit) random projections of the latent variables as a way to define the covariance matrices used in the Variational approximations adopted in the context of Bayesian Neural Networks.

We focus on the application of Random Projection to Variational Auto-Encoders. While the authors are unaware of prior work on this subject, numerous other extensions to Variational Auto-Encoders (VAEs) have been proposed. For example: Eastwood & Williams (2018) and Chen et al. (2018) consider variants of the

VAE focusing on disentangling representations; Janjoš et al. (2023) and Vahdat & Kautz (2020) use the Unscented transform to improve on the performance of VAEs; Tolstikhin et al. (2017) proposes the use of the Wasserstein distance to define its loss function. We perceive these papers describe potentially synergistic advances with the approach described herein.

There are various ways to improve the latent space and align it with the posterior distribution space. Normalising flows stands out as a leading technique acknowledged for its ability to transform the latent spaces into complex distributions (Kobyzev et al., 2020). Normalising flows has proven to be effective in the context of variational inference (Rezende & Mohamed, 2016) and in extension of VAEs as well (Mouton & Kroon, 2023). In a study by Luo & Chien (2021), they addressed the focal issue as having simple isotropic Gaussian priors as base assumptions for latent variables, is insufficient to ensure the desired diversity of generated responses. While tackling that specific issue Normalising flows are still considered practically restrictive while performing variational inference as they require analytically invertible flows (Giaquinto & Banerjee, 2020). Extending their own work on VAEs, Kingma et al. (2016) proposed the use of autoregressive neural networks to structure a chain of invertible transformations to create a flow of a flexible distribution. Different variations of Normalising flows have achieved a more adaptable distribution flow with the state-of-the-art focusing on autoregressive Neural Networks (Ziegler & Rush, 2019), (Papamakarios et al., 2017).

We stray away from prevailing methodologies that rely on Neural Networks to tackle the intricate task of approximating the complex latent distribution. We follow the ideology of Dai & Wipf (2019) and contrast the common belief that Gaussian approximations are inferior to their NN counterparts. The use of Random Projection can benefit existing works with insights achieved from normalising flows (Kobyzev et al., 2019) with as little computational burden as possible. As highlighted in Giaquinto & Banerjee (2020), the introduction of heavy complexity in approaches like Papamakarios et al. (2017) was used to generate an appropriate flow for a complicated latent distribution. On the other hand, Random projection does not limit the use of normalising flows. A strategic application of Random Projection could complement works such as Kingma et al. (2016). Kingma et al. (2016) which have adopted a diagonal parameterisation of their covariance alongside Normalising flows to achieve their results within a reasonable timeframe.

## 3 Methodology

### 3.1 General Setting

We aim to align our description with the notation in Kingma & Welling (2013) and refer the reader to that paper for further details. Here, we briefly summarise the problem we consider such that the context of our contribution is clear. We begin by stating that we consider a dataset, $\mathbf{X} = \left\{\mathbf{x}^{(i)}\right\}_{i=1}^{N}$, comprising $N$ i.i.d. samples of $\mathbf{x} \in \mathbb{R}^{D}$. We assume that each datum is generated by sampling an unobserved 'latent' variable, $\mathbf{z}^{(i)} \in \mathbb{R}^{J}$ (where we assume $J \ll D$), from a prior, $p_\theta(\mathbf{z})$, and then sampling the datum, $\mathbf{x}^{(i)}$, from the likelihood, $p_\theta\left(\mathbf{x}|\mathbf{z}^{(i)}\right)$. Note that the prior and likelihood are parameterised by $\theta$, which we wish to optimise such that we can have an expressive model that can compress the information in the data into the (lower-dimensional) latent space.

Since the posterior for each datum, $p_\theta\left(\mathbf{z}|\mathbf{x}^{(i)}\right)$ is, in general, intractable, we introduce a variational approximation, $q_\phi\left(\mathbf{z}|\mathbf{x}^{(i)}\right)$. This variational approximation is a member of a parametric family and is parameterised by $\phi$. We wish to optimise $\phi$ such that we can obtain an accurate approximation to the posterior.

We choose to optimise a lower bound on the log-likelihood of the data, marginalised over the latent variables. We note that, since the data are i.i.d., the log-likelihood of the data is just a sum (over the data) of each datum's individual (marginal) log-likelihood. We can calculate the lower bound on the log-likelihood of the $i$th datum as

$$\log p_\theta\left(\mathbf{x}^{(i)}\right) \geq \mathcal{L}\left(\theta, \phi; \mathbf{x}^{(i)}\right) = \mathbb{E}_{q_\phi\left(\mathbf{z}|\mathbf{x}^{(i)}\right)}\left[\frac{p_\theta\left(\mathbf{x}^{(i)}, \mathbf{z}\right)}{q_\phi\left(\mathbf{z}|\mathbf{x}^{(i)}\right)}\right] \tag{1}$$

where we can decompose this bound into two terms: a negative Kullback–Leibler (KL) divergence (between the variational approximation to the posterior and the prior) and the expected reconstruction error,

$$\mathcal{L}\left(\theta, \phi; \mathbf{x}^{(i)}\right) = - D_{\mathrm{KL}}\left(q_\phi\left(\mathbf{z}|\mathbf{x}^{(i)}\right) \| p_\theta\left(\mathbf{z}\right)\right)$$
$$+ \mathbb{E}_{q_\phi\left(\mathbf{z}|\mathbf{x}^{(i)}\right)}\left[\log p_\theta\left(\mathbf{x}^{(i)}|\mathbf{z}\right)\right] \quad (2)$$

We approximate the expected reconstruction error using the reparameterization trick such that we can optimise the lower bound on the log-likelihood using popular stochastic optimisation methods. This involves approximating the expected reconstruction error as follows:

$$\mathbb{E}_{q_\phi\left(\mathbf{z}|\mathbf{x}^{(i)}\right)}\left[\log p_\theta\left(\mathbf{x}^{(i)}|\mathbf{z}\right)\right] \approx \frac{1}{L}\sum_{l=1}^{L}\log p_\theta\left(\mathbf{x}^{(i)}|\mathbf{z}^{(i,l)}\right) \quad (3)$$

where $\mathbf{z}^{(i,l)} = g_\phi\left(\epsilon^{(i,l)}, \mathbf{x}^{(i)}\right)$, $g_\phi\left(.\right)$ is a function parameterised by $\phi$ (so to be optimised) and $\epsilon^{(i,l)} \sim p\left(\epsilon\right)$.

### 3.2 Specifics of Variational Auto-Encoders

We consider the case where the prior on the latent space is an isotropic multivariate Gaussian with no dependence on $\theta$, $p_\theta\left(\mathbf{z}\right) = \mathcal{N}\left(\mathbf{z}; \mathbf{0}_J, \mathbf{I}_J\right)$, where $\mathbf{0}_J$ is a vector comprising $J$ zeros and $\mathbf{I}_J$ is the $J \times J$ Identity matrix. We consider a likelihood $p_\theta\left(\mathbf{x}|\mathbf{z}\right)$ that is a Gaussian MLP. We also assume that the variational approximation to the posterior for each datum is Gaussian, $q_\phi\left(\mathbf{z}|\mathbf{x}^{(i)}\right) = \mathcal{N}\left(\mathbf{z}; \mu^{(i)}, \mathbf{\Sigma}^{(i)}\right)$

### 3.3 Alternative Parameterisations of the Covariance Matrices

The focus of this paper is on how we parameterise $\mathbf{\Sigma}^{(i)}$. To ensure $\mathbf{\Sigma}^{(i)}$ is positive semi-definite while also enabling an optimiser to consider an unconstrained space, we begin by noting that $\mathbf{\Sigma}^{(i)} = \mathbf{\Sigma}^{(i)\frac{1}{2}}\mathbf{\Sigma}^{(i)\frac{1}{2}T}$. We note that this is similar but distinct from considering a Cholesky decomposition since we do not impose the constraint that the diagonals entries of $\mathbf{\Sigma}^{(i)\frac{1}{2}}$ are positive.

We then consider three parameterisations and associated denotations,

$$\mathbf{\Sigma}^{(i)\frac{1}{2}} = \begin{cases} \mathbf{Q}_J^{(i)} & \text{Full rank, 'Full',} \\ \mathbf{\Lambda}_{J,n}^{(i)}\mathbf{Q}_n^{(i)} & \text{Random Projection, 'RP',} \\ \sigma^{(i)} & \text{Diagonal, 'D',} \end{cases} \quad (4)$$

where $\mathbf{Q}_d^{(i)}$ is a lower triangular $d \times d$ matrix (for $d \in \{J, n\}$), $\mathbf{\Lambda}_{J,n}^{(i)}$ is a matrix that projects from $n < J$ dimensions to $J$ dimensions and $\sigma^{(i)}$ is a diagonal $J \times J$ matrix. Note that, in the software implementation of D, the diagonal entries of $\sigma^{(i)}$ are stored on a log scale.

### 3.4 Implementation of Random Projection

We argue that random projection produces a more sophisticated approach to dimensionality reduction relative to simply constraining the matrix to be diagonal. We consider the random projection in the same way as the reparameterised error input is considered in the approximation to the expected reconstruction error: we sample the projection for each datum and then consider this to be fixed for the duration of the optimisation. We then optimise the remaining parameters of each covariance matrix.

To construct a random projection from $n$ to $J$ dimensions, we sample $n$ vectors in $J$ dimensions from an isotropic Normal distribution and then use QR decomposition (Anderson et al., 1992) to construct an orthonormal basis, which we consider to be the random projection. We store the projection matrices in memory so as to not generate them at each iteration of the training. This speeds up the runtime execution at the expense of sacrificing memory. In the contexts we have considered, the benefit in terms of speed

outweighs the impact on memory usage. However, we note that, in other settings, it may be preferable to recalculate the projection matrices at each iteration of the training.

The number of parameters per datum used to parameterise the covariance matrices is then as described in table 1. Note that by choosing appropriate values for $n$, the RP parameterisation can be configured to have parameter counts between those associated with a full covariance matrix and a diagonal covariance matrix.

Table 1: Number Of Optimised Parameters Per Datum For Each Parameterisation

| Parameterisation | Parameter count |
| --- | --- |
| Full | $\frac{1}{2} \times J \times (J+1)$ |
| RP | $\frac{1}{2} \times n \times (n+1)$ |
| D | $J$ |

### 3.5 KL Divergence Terms

As is described in more detail in the supplementary material, in the case of a generic covariance matrix, $\boldsymbol{\Sigma}^{(i)}$, the KL divergence term can be expressed as follows:

$$D_{\mathrm{KL}} \left( q_\phi \left( \mathbf{z}|\mathbf{x}^{(i)} \right) \parallel p_\theta \left( \mathbf{z} \right) \right)$$
$$= \frac{1}{2} \left( \log |\boldsymbol{\Sigma}^{(i)}| + J - \mathrm{Tr} \left[ \boldsymbol{\Sigma}^{(i)} \right] - \sum_{j=1}^{J} \left( \mu_j^{(i)} \right)^2 \right) \tag{5}$$

where $\mu_j^{(i)}$ is the $j$th element of $\mu^{(i)}$.

For each of the alternative parameterisations we consider, we can calculate the KL divergence term as follows:

$$\log |\boldsymbol{\Sigma}^{(i)}| = \begin{cases} \log |\mathbf{Q}_J^{(i)} \mathbf{Q}_J^{(i)^T}| & \text{Full} \\ \log |\mathbf{Q}_n^{(i)} \mathbf{Q}_n^{(i)^T}| & \text{RP} \\ \sum_{j=1}^{J} \log \left( \left( \sigma_{j,j}^{(i)} \right)^2 \right) & \text{D} \end{cases} \tag{6}$$

where $\sigma_{j,j}^{(i)}$ is the $j$th diagonal entry of $\sigma^{(i)}$ and

$$\mathrm{Tr} \left[ \boldsymbol{\Sigma}^{(i)} \right] = \begin{cases} \mathrm{Tr} \left[ \mathbf{Q}_J^{(i)} \mathbf{Q}_J^{(i)^T} \right] & \text{Full} \\ \mathrm{Tr} \left[ \boldsymbol{\Lambda}_{J,n}^{(i)} \mathbf{Q}_n^{(i)} \left( \boldsymbol{\Lambda}_{J,n}^{(i)} \mathbf{Q}_n^{(i)} \right)^T \right] & \text{RP} \\ \sum_{j=1}^{J} \left( \sigma_{j,j}^{(i)} \right)^2 & \text{D} \end{cases} \tag{7}$$

where the simplifications used are derived in the supplementary material A, , and the expressions in Kingma & Welling (2013) are equivalent to those denoted diagonal (D) herein.

## 4 Experiments

The focus of our experiments is on understanding the utility of different parameterisations of the covariances matrices used in the variational approximation: we aim to understand whether any reduction in loss (and both its constituents, KL divergence and expected reconstruction loss) is coupled to any increase in computational time. We consider the three parameterisations described above: Full; D (as described in Kingma & Welling

(2013)); and RP. Creating a middle ground with the RP parameterisation between D and Full, we examine our intuition regarding the correlations' impact on expressing a latent space with various volumes of $J$. We do not use Full, which could be considering spurious correlations, or D, which assumes that all variables are independent within the covariance.

We compare performance in the context of learning a latent space to describe MNIST, CIFAR-10 and Flowers-102. We use the same network topology as is considered in Kingma & Welling (2013) (comprising three convolutional layers in both the encoder and decoder) along with a batch size of 100, a learning rate of $3 \times 10^{-5}$, a weight decay of $10^{-6}$ and 500 hidden units for the MNIST dataset with 60,000 training and 10,000 test cases. We then use the same topology but with 750 hidden units for the CIFAR-10 dataset with 50,000 training and 10,000 test cases. Finally, we increased the number of hidden units to 1000 for Flowers-102 with 6149 training and 1020 test cases.

We report the training and testing loss (i.e., the lower bound on the log-likelihood) as well as its constituent terms: negative KL divergence and expected reconstruction error. We also report run time in the context of processing training cases using an implementation in Python and a machine with an NVIDIA P100 GPU and 10 Intel Xeon Gold Skylake cores of 9.6 GB of memory per core. We train each algorithm once, but since the VAE's behaviour is stochastic given the inputs, then quantify the statistical fluctuations in performance that result from 10 Monte Carlo runs per datum. We used paired statistical tests to minimise the resulting Monte Carlo variance, i.e., we assess all approaches using the same portfolio of random number seeds.

We consider a sequence of experiments to understand: the loss associated with using different parameterisations (and how this varies with $J$ and with $n$ for RP); the run times associated with using parameterisations (and how this varies with $n$); how the loss is decomposed into its constituent terms and whether there is any evidence of overfitting; qualitative performance.

### 4.1 Total Loss

We begin by considering the total loss for each of the parameterisations in the context of each of the three datasets. Tables 2, 3 and 4 respectively consider the MNIST and CIFAR-10 datasets. In each case, we compare the total loss for each parameterisation as a function of $J$. In the case of the RP, we consider the variable $n$ as a subset of integer values derived from the variable $J$. Specifically, $n$ encompasses five distinct values: 10%, 30%, 50%, 70%, and 100% of the value of $J$. We present a subsample of the distinct $n$ values according to the dataset. Moreover, we include the standard deviation of 5 Monte Carlo test runs in tables 2, 3 and 3. In bold, we highlight the lowest Total test loss regardless of its volatility, which is discussed later in 5.

Table 2: Total Testing Loss For Each Parameterisation For MNIST

| Parameterisation | J=10 | J=30 | J=50 | J=70 | J=100 |
|---|---|---|---|---|---|
| Full | 162.42±31.55 | 161.79±26.28 | 193.34±13.18 | 187.34±22.37 | 168.32±5.39 |
| D | 108.47±0.17 | 102.07±0.55 | 103.27±0.40 | **103.56**±0.41 | 102.84±0.62 |
| RP ($n = 10\%$) | **94.00**±0.59 | **85.17**±8.94 | **80.83**±2.52 | 105.92±11.55 | **83.57**±6.15 |
| RP ($n = 30\%$) | 107.27±3.30 | 115.66±5.20 | 101.85±5.95 | 122.25±14.08 | 109.50±4.87 |
| RP ($n = 50\%$) | 120.57±4.73 | 128.16±15.06 | 121.77±5.76 | 118.82±4.44 | 134.74±2.13 |
| RP ($n = 70\%$) | 135.55±6.78 | 127.45±9.26 | 132.14±3.10 | 138.64±2.44 | 148.00±0.76 |
| RP ($n = J$) | 137.21±6.49 | 145.88±6.16 | 143.41±7.33 | 172.32±5.32 | 171.96±2.63 |

Table 3: Total Testing Loss For Each Parameterisation For CIFAR-10

| Parameterisation | J=10 | J=50 | J=70 | J=100 |
|---|---|---|---|---|
| Full | 1931.21±55.94 | 1988.16±34.90 | 2396.98±83.90 | 2099.05±4.41 |
| D | 1839.36±0.55 | 1794.64±0.35 | **1792.86**±0.39 | 1793.73±0.40 |
| RP ($n=10\%$) | **1824.86**±2.42 | **1790.51**±15.93 | 1795.73±22.19 | **1762.95**±9.22 |
| RP ($n=30\%$) | 1850.82±7.94 | 1822.15±13.83 | 1837.52±9.99 | 2021.14±50.50 |
| RP ($n=50\%$) | 1904.15±22.46 | 1979.97±85.03 | 1972.00±19.57 | 2058.60±12.96 |
| RP ($n=70\%$) | 1917.47±26.10 | 1873.65±8.81 | 2103.59±11.94 | 1995.11±17.09 |
| RP ($n=J$) | 1887.06±13.79 | 2023.74±5.62 | 2134.38±9.21 | 2260.81±10.49 |

Table 4: Total Testing Loss For Each Parameterisation For Flowers-102

| Parameterisation | J=50 | J=100 | J=150 |
|---|---|---|---|
| Full | 6762.11±10.19 | 6734.63±21.41 | 6784.30±21.42 |
| D | 6803.96±4.03 | 6666.73±3.81 | 6616.18±2.69 |
| RP ($n=10\%$) | **6659.60**±0.75 | **6539.42**±11.04 | **6484.76**±7.65 |
| RP ($n=30\%$) | 6711.32±2.81 | 6559.33±3.20 | 6587.78±6.55 |
| RP ($n=50\%$) | 6682.83±4.51 | 6675.89±8.48 | 6589.11±5.45 |
| RP ($n=70\%$) | 6718.75±3.23 | 6659.12±6.63 | 6677.32±5.19 |
| RP ($n=J$) | 6764.82±1.77 | 6738.99±5.51 | 6763.29±4.39 |

As a more visual comparison of the expressive ability of RP with $1 < n \leq J$, we provide the total Test loss of the two extremes of the RP parameterisation, where $n = 10\%$ and $n = J$, and for $n = 50\%$ for the three datasets in Figures 1, 2, 3

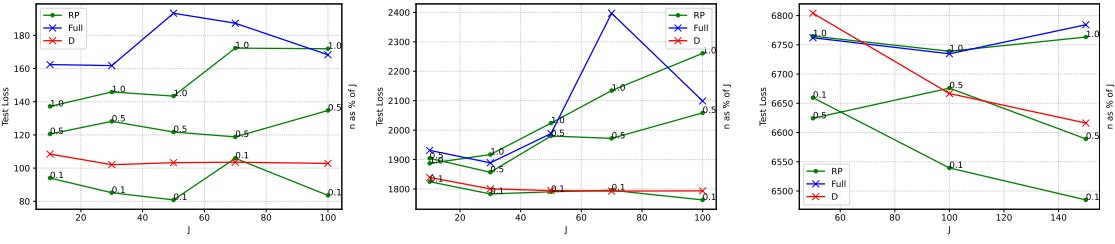

Figure 1: MNIST        Figure 2: CIFAR-10        Figure 3: Flowers-102

## 4.2   Run time

First and foremost, we are interested in how much computational overhead the RP parameterisation adds. We can gauge this by comparing the time required for fundamental operations, such as matrix inversion, across the different parameterisations D, Full, and RP while also considering the projection function that is required for RP. In fig 4, we illustrate the time complexity of D, Full, and RP with various $n$. It's evident that for smaller $n$, RP initially offers faster computations. However, as $n$ increases, RP becomes notably more complex compared to the Full rank covariance. When $n = J$, RP has the same complexity for the matrix inversion with the Full parameterisation; however, we have extra computational time as we need to calculate the projection of the covariance.

We now consider the computational time, as quantified by the minimum runtime out among 10 samples of 100 batch iterations, for each of the parameterisations in the context of each of the three datasets. Figures 5, 6 and 7 present the runtime per batch iteration, with increments ranging from $1 < n \leq J$.

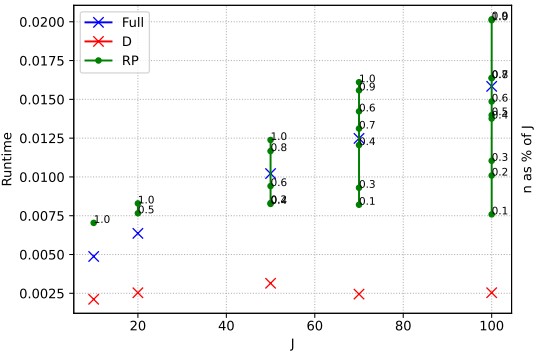 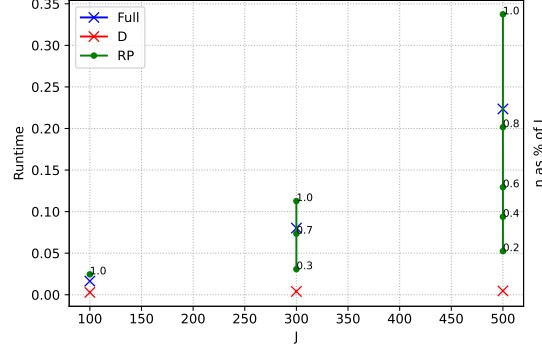

Figure 4: Runtime of expected complexity with different parameterisations for varying $J$

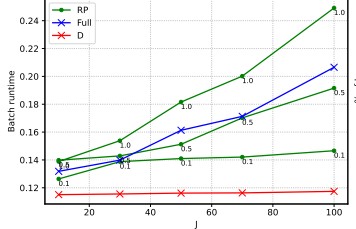 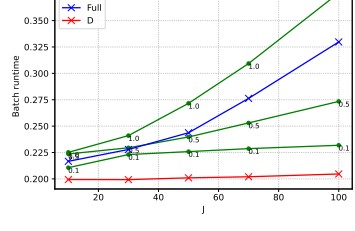 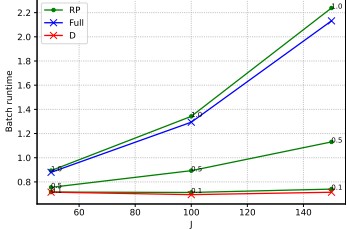

Figure 5: MNIST          Figure 6: CIFAR-10          Figure 7: Flowers-102

## 4.3 Constituents of Loss

We then focus on the Flowers102 dataset and consider the decomposition of the loss into its constituent terms. Table 5 describes how the loss is decomposed into the (negative) KL divergence and expected reconstruction error.

Highlighted in bold for both tables are the equivalent values from Averaged Total loss table 4.

Table 5: Reconstruction Error And KL Divergence (Total Loss) For Flowers-102: Reconstruction Error Listed First.

| Parameterisation | J=50 | J=100 | J=150 |
|---|---|---|---|
| Full | 6637.04/15.24 | 6532.98/189.43 | 6476.39/305.67 |
| D | 6704.65/91.64 | 6507.21/157.89 | 6449.93/167.59 |
| RP ($n = 10\%$) | **6633.32/26.05** | **6501.35/37.50** | **6431.76/48.32** |
| RP ($n = 30\%$) | 6656.87/51.67 | 6482.34/72.61 | 6464.49/127.15 |
| RP ($n = 50\%$) | 6606.02/70.89 | 6656.21/122.51 | 6435.96/149.06 |
| RP ($n = 70\%$) | 6623.28/92.03 | 6678.24/167.36 | 6462.02/219.30 |
| RP ($n = J$) | 6640.61/124.07 | 6685.93/222.22 | 6469.83/293.24 |

## 4.4 Qualitative Performance

Finally, we compare qualitative performance by visualising the CIFAR-10 dataset using the D and RP parameterisations: for 16 test images, we encode the images in the latent space and then (stochastically) decode the latent encodings of each image. See figure 8. Further examples are provided in the supplementary material.

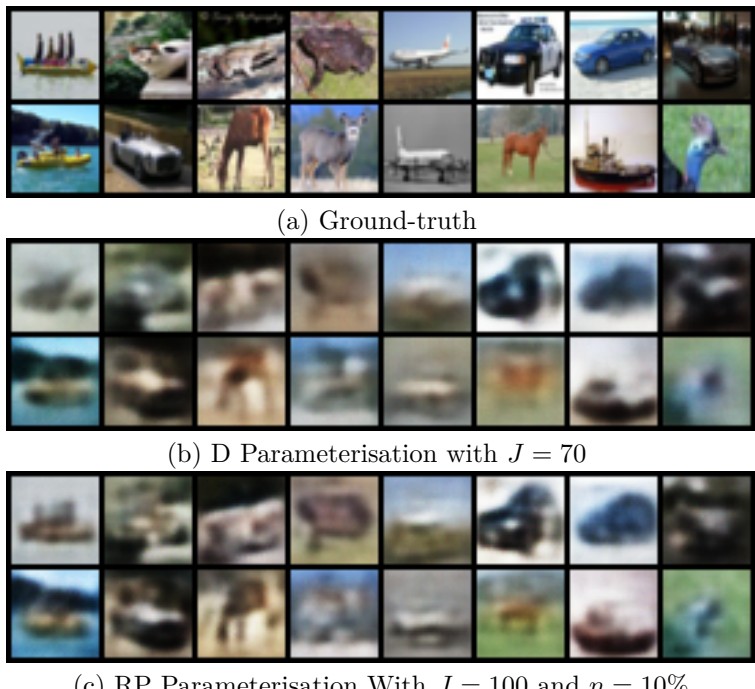

(a) Ground-truth

(b) D Parameterisation with $J = 70$

(c) RP Parameterisation With $J = 100$ and $n = 10\%$

Figure 8: CIFAR10 D/RP parameterisation with the lowest Test losses

## 5    Discussion

We first want to outline the standard deviation of the 5 Monte Carlo (MC) runs. Considering the stochasticity of the algorithm, for all test results in tables 2 and 3, the D parameterisation produces more constant results compared to the others (according to the standard deviation of the multiple test runs). Although the RP parameterisation has a lower total loss, we observe a higher MC standard deviation due to also accounting for the correlations of the latent space covariance. Especially on a few occasions, the other parameterisations produce greatly more volatile results than D. Interestingly enough, that can directly be compared with the expressivity of the parameterisations as we have a range of parameter spaces to optimise instead of the minimum or maximum volume ( 1). With a Full parameterisation of $J(J + 1)/2$ variables, there are more variables to optimise compared to the D parameterisation ($Jx1$), as well as a different structure, highlighting the significance of considering the covariance's correlations. Variations in standard deviation for RP are evident. We notice a slight shift of consistent generations in table 4 where the RP parameterisation ($J = 50$,$N = 10\%$) produces a lower MC standard deviation for a more plentiful feature space ($3 \times 64 \times 64$). Overall, it becomes apparent that parameterisation expressivity significantly influences the mean MC Total loss of tables 2, 3 and 4. RP consistently outperforms other parameterisations with a lower variable space $n$, apart from two cases of $J = 70$ for MNIST and CIFAR-10 without, of course, factoring in the lowest attainable value for RP wrt the MC standard deviation. Given this observation, we observe that $n^\star$, the optimal value of $n$ for minimising the loss, always appears to be such that $1 < n^\star \ll J$, at least for the limited number of iterations we performed. This implies a balance between having sufficiently many parameters to characterise the uncertainty and sufficiently few parameters to avoid overfitting (and therefore sufficiently few parameters to avoid taking spurious correlations into account). We also note that the optimal value of $J$ and $n$, and so the inherent dimensionality necessary for the latent space to capture the uncertainty and correlations of the data, varies between algorithms and dataset. Again, this implies the need to strike a problem-specific balance between expressiveness and the potential for overfitting.

We also visually showcase the ability to express a space in between Full and D with RP through figures 1, 2 and 3. Remarkably, we notice that for $n = 10\%$ of $J$ we RP has a lower or at least commensurate total loss with D. It is more important for a covariance to consider correlations instead of assuming independent

variances for its samples. On the other hand, for all cases, the Full parameterisation Total loss remains higher than the rest, with RP following the same trend when $n = J$ (i.e. $n = 100\%$). Per our expectations, the ratio of spurious correlations is highest when considering Full, and RP $n \leftarrow J$ yields higher losses. The most intriguing observation derived from 3 is that when increasing $J$, the D's parameterisation loss is higher than Full for $J = 50$ and has a decreasing tendency towards $J = 150$ with the RP parameterisation following the same decreasing trend and remaining lower than D's total loss. Indicating that the latent space requires more than $150 \times 1$ parameters and less than $(50 \times (50 + 1)/2)$ to be expressed. We expect the RP loss to increase over D's when the volume of spurious correlations begins to become redundant as $n \leftarrow J$ or $J$ increases, as can be seen even for $n = 10\%$ of $J = 70$ for the other two datasets 1 and 2.

Figures 5, 6, 7 make clear that, for the values of $n$ considered, using RP does incur a computational cost, but that this computational cost as expected ( 4) for smaller $n$ the impact is negligible. The results of the Flowers-102 dataset ( 7) follow the expected runtime load of the higher $J$ dimensions. The computational overhead of manipulating the random projections at each iteration is not as insignificant as one might expect when $n \to J$; setting $n = 100\%$ results in the greatest computational cost in all cases.

Examining table 5, it becomes evident that both the KL divergence and expected reconstruction error follow the trend observed in the MC mean Total loss presented in table 4. With a much lower KL divergence loss, the RP parameterisation minimises the Total loss. Moreover, with lower KL divergence describing a better approximation of the latent space, the reconstruction error of the VAE decreases, also assisting in lowering the Total loss.

Finally, we generate visualisations for the CIFAR-10 dataset, each one being visually distinct from the others due to the wide variety of cases represented within the test cases. From inspecting the figures 8, we judge that the resolution achieved using RP is superior to that achieved using D, i.e., achieved when using Kingma & Welling (2013). We notice that the massively varying examples of CIFAR-10 with RP are visually distinguishable from the generations obtained from D. See also supplementary material for more example visualisations of the other datasets (available in appendix B).

# 6 Conclusion

We consider alternative parameterisation of the covariance matrix used in a Variational Auto-Encoder's approximation to each datum's posterior in the latent space. We specifically consider a parameterisation based on a random projection from a lower-dimensional space in which the covariance is full rank. We find this parameterisation provides a useful compromise with respect to both KL divergence and reconstruction error: in the context of MNIST, CIFAR-10 and Flowers-102, this approach is shown to offer an improved ability to avoid overfitting while allowing an expressive description of the posterior. We find that the computational cost of this approach is not prohibitive. A full rank covariance would capture the correlation of the desired latent space with enough training iterations, but we can approximate those relations by projecting a smaller full rank covariance with the RP parameterisation a lot faster.

# 7 Future Work

In future work, we hope to improve the performance of the approach by considering the projections to be parameters that can be optimised rather than random. Given that the random projection introduces another hyperparameter, the dimensionality of the low-dimensional space, we also hope to develop techniques to automate the setting of this (and other) hyperparameters.

Future work could also consider the wider application of the novel variants of Variational Auto-Encoders described in this paper. In particular, the authors perceive it would be interesting to assess the benefits of the approach in the context of other applications involving complex correlation structures, e.g. stock market data.

More generally, given the utility of the approach in the context of Variational Auto-Encoders, future work would sensibly consider the application of random projections both in the context of Variational approaches

to other variants of Variational Auto-Encoders with and without Normalising flows to examine the synergy of the two methods as well as training of other Bayesian Neural Network models.

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

## A  Derivation of KL Divergence Terms

We now present the derivation of the KL divergence terms stated in section 3.5, noting that the content herein is a mild generalisation of Kingma & Welling (2013).

We begin by noting that the KL divergence is:

$$D_{\mathrm{KL}}\left(q_\phi\left(\mathbf{z}|\mathbf{x}^{(i)}\right) \parallel p_\theta\left(\mathbf{z}\right)\right) = \int q_\phi\left(\mathbf{z}|\mathbf{x}^{(i)}\right) \log\left[\frac{q_\phi\left(\mathbf{z}|\mathbf{x}^{(i)}\right)}{p_\theta\left(\mathbf{z}\right)}\right] d\mathbf{z} \tag{8}$$

$$= \int q_\phi\left(\mathbf{z}|\mathbf{x}^{(i)}\right) \log q_\phi\left(\mathbf{z}|\mathbf{x}^{(i)}\right) d\mathbf{z} - \int q_\phi\left(\mathbf{z}|\mathbf{x}^{(i)}\right) \log p_\theta\left(\mathbf{z}\right) d\mathbf{z} \tag{9}$$

Recall that we consider that

$$q_\phi\left(\mathbf{z}|\mathbf{x}^{(i)}\right) = \mathcal{N}\left(\mathbf{z}; \mu^{(i)}, \mathbf{\Sigma}^{(i)}\right) \tag{10}$$

$$p_\theta\left(\mathbf{z}\right) = \mathcal{N}\left(\mathbf{z}; \mathbf{0}_J, \mathbf{I}_J\right) \tag{11}$$

which are such that

$$\log q_\phi\left(\mathbf{z}|\mathbf{x}^{(i)}\right) = -\frac{J}{2}\log 2\pi - \frac{1}{2}\log\left|\mathbf{\Sigma}^{(i)}\right| - \frac{1}{2}\left(\mathbf{z} - \mu^{(i)}\right)^T \mathbf{\Sigma}^{(i)-1}\left(\mathbf{z} - \mu^{(i)}\right) \tag{12}$$

$$\log p_\theta\left(\mathbf{z}\right) = -\frac{J}{2}\log 2\pi - \frac{1}{2}\mathbf{z}^T\mathbf{z} \tag{13}$$

and so

$$\int q_\phi\left(\mathbf{z}|\mathbf{x}^{(i)}\right) \log q_\phi\left(\mathbf{z}|\mathbf{x}^{(i)}\right) d\mathbf{z} = -\frac{J}{2}\log 2\pi - \frac{1}{2}\log\left|\mathbf{\Sigma}^{(i)}\right| - \frac{J}{2} \tag{14}$$

$$\int q_\phi\left(\mathbf{z}|\mathbf{x}^{(i)}\right) \log p_\theta\left(\mathbf{z}\right) d\mathbf{z} = -\frac{J}{2}\log 2\pi - \frac{1}{2}\mathrm{Tr}\left[\mathbf{\Sigma}^{(i)}\right] - \frac{1}{2}\mu^{(i)T}\mu^{(i)} \tag{15}$$

$$= -\frac{J}{2}\log 2\pi - \frac{1}{2}\mathrm{Tr}\left[\mathbf{\Sigma}^{(i)}\right] - \frac{1}{2}\sum_{j=1}^{J}\left(\mu_j^{(i)}\right)^2 \tag{16}$$

where $\mathrm{Tr}\left[A\right]$ is the trace of $A$ such that

$$D_{\mathrm{KL}}\left(q_\phi\left(\mathbf{z}|\mathbf{x}^{(i)}\right) \parallel p_\theta\left(\mathbf{z}\right)\right) = \frac{1}{2}\left(\log\left|\mathbf{\Sigma}^{(i)}\right| + J - \mathrm{Tr}\left[\mathbf{\Sigma}^{(i)}\right] - \sum_{j=1}^{J}\left(\mu_j^{(i)}\right)^2\right) \tag{17}$$

which is identical to (5). To derive (15), we note that, by definition,

$$\mathbf{\Sigma}^{(i)} = \int \mathcal{N}\left(\mathbf{z}; \mu^{(i)}, \mathbf{\Sigma}^{(i)}\right)\left(\mathbf{z} - \mu^{(i)}\right)^T\left(\mathbf{z} - \mu^{(i)}\right) d\mathbf{z} \tag{18}$$

$$= \int \mathcal{N}\left(\mathbf{z}; \mu^{(i)}, \mathbf{\Sigma}^{(i)}\right) \mathbf{z}^T\mathbf{z} d\mathbf{z} - \mu^{(i)T}\mu^{(i)} \tag{19}$$

and so

$$\int \mathcal{N}\left(\mathbf{z};\mu^{(i)},\mathbf{\Sigma}^{(i)}\right)\mathbf{z}^T\mathbf{z}d\mathbf{z} = \int \mathcal{N}\left(z;\mu^{(i)},\mathbf{\Sigma}^{(i)}\right)\mathrm{Tr}\left[\mathbf{z}\mathbf{z}^T\right]d\mathbf{z} \tag{20}$$

$$=\mathrm{Tr}\left[\int \mathcal{N}\left(\mathbf{z};\mu^{(i)},\mathbf{\Sigma}^{(i)}\right)\mathbf{z}\mathbf{z}^T d\mathbf{z}\right] \tag{21}$$

$$=\mathrm{Tr}\left[\mathbf{\Sigma}^{(i)} + \mu^{(i)}\mu^{(i)T}\right] \tag{22}$$

$$=\mathrm{Tr}\left[\mathbf{\Sigma}^{(i)}\right] + \sum_{j=1}^{J}\left(\mu_j^{(i)}\right)^2. \tag{23}$$

### A.1 Specific Parameterisations

We now consider the specific parameterisations of the covariance, $\mathbf{\Sigma}^{(i)}$, defined in (4) and derive the simplified expressions that are stated in (6) and (7). We assume the expressions for the full covariances are self-evident, but, for completeness, do consider the expressions for the diagonal covariance, even though this is repeating information provided in Kingma & Welling (2013).

#### A.1.1 Random Projection, RP

We have constructed $\mathbf{\Lambda}_{J,n}^{(i)}$ to be a projection comprising an orthonormal basis such that

$$\log|\mathbf{\Sigma}^{(i)}| = \log\left|\mathbf{\Lambda}_{J,n}^{(i)}\mathbf{Q}_n^{(i)}\left(\mathbf{\Lambda}_{J,n}^{(i)}\mathbf{Q}_n^{(i)}\right)^T\right| = \log\left|\mathbf{\Lambda}_{J,n}^{(i)}\mathbf{Q}_n^{(i)}\mathbf{Q}_n^{(i)T}\mathbf{\Lambda}_{J,n}^{(i)T}\right| \tag{24}$$

$$=\log\left|\mathbf{Q}_n^{(i)}\mathbf{Q}_n^{(i)T}\right| \tag{25}$$

where we note that no simplification is possible when calculating the trace as

$$\mathrm{Tr}\left[\mathbf{\Sigma}^{(i)}\right] = \mathrm{Tr}\left[\mathbf{\Lambda}_{J,n}^{(i)}\mathbf{Q}_n^{(i)}\left(\mathbf{\Lambda}_{J,n}^{(i)}\mathbf{Q}_n^{(i)}\right)^T\right]. \tag{26}$$

#### A.1.2 Diagonal, D

Since $\mathbf{\Sigma}^{(i)}$ is diagonal, its determinant and trace can be expressed in terms of its diagonal entries as

$$\log|\mathbf{\Sigma}^{(i)}| = \log|\sigma^{(i)}\sigma^{(i)T}| = \sum_{j=1}^{J}\log\left(\left(\sigma_{j,j}^{(i)}\right)^2\right) \tag{27}$$

$$\mathrm{Tr}\left[\mathbf{\Sigma}^{(i)}\right] = \mathrm{Tr}\left[\sigma^{(i)}\sigma^{(i)T}\right] = \sum_{j=1}^{J}\left(\sigma_{j,j}^{(i)}\right)^2. \tag{28}$$

## B Additional Results

We now present some additional qualitative test results for both datasets using each of the parameterisations and as a function of the dimensionality of the latent space, $J$ and the dimensionality of the low-dimensional space considered in the random projection, $n$. For the MNIST dataset we visualise the best (lowest total test loss) test cases to contrast the resolution clarity of the ground-truth (shown in figure 9) and the results obtained using each of the parameterisations: Full, D, RP and RP-D, respectively shown in figures 10, 11 and 12. We vary $J$ and consider $n = \frac{J}{10}$ throughout. For the Flowers-102 dataset, we present the ground truth (in figure 13) along with the test cases using the values of $J$ that minimised the total loss. For the results for the Full parameterisation shown in figure 14, we therefore consider $J = 50$, for the results of the D parameterisation we show $J = 150$ (figure 15). For the RP parameterisation, we provide the best cases of n in figures 16.

Our view is that qualitative inspection of these images (from the datasets) validates the relative quantitative performance reported in section 4 and discussed in section 5.

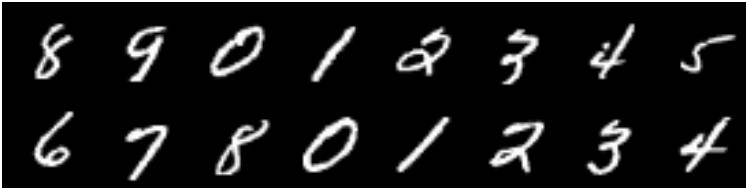

Figure 9: MNIST Test Ground-truth

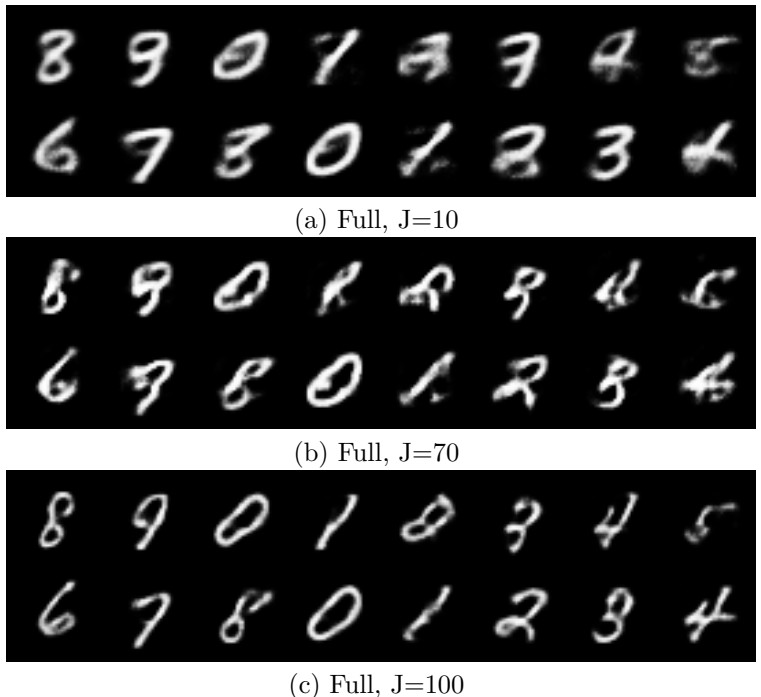

(a) Full, J=10

(b) Full, J=70

(c) Full, J=100

Figure 10: Stochastic Visualisation Of MNIST (Full Parameterisation)

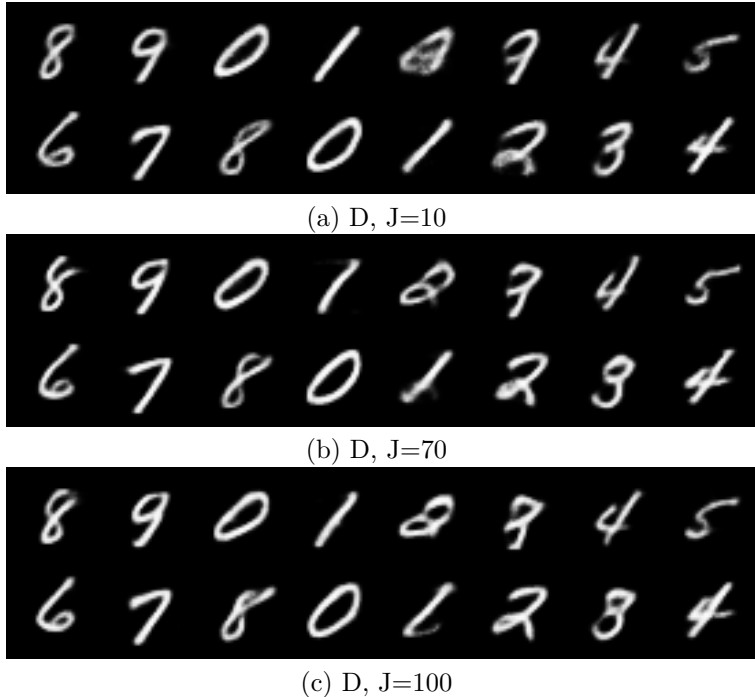

(a) D, J=10

(b) D, J=70

(c) D, J=100

Figure 11: Stochastic Visualisation Of MNIST (D Parameterisation)

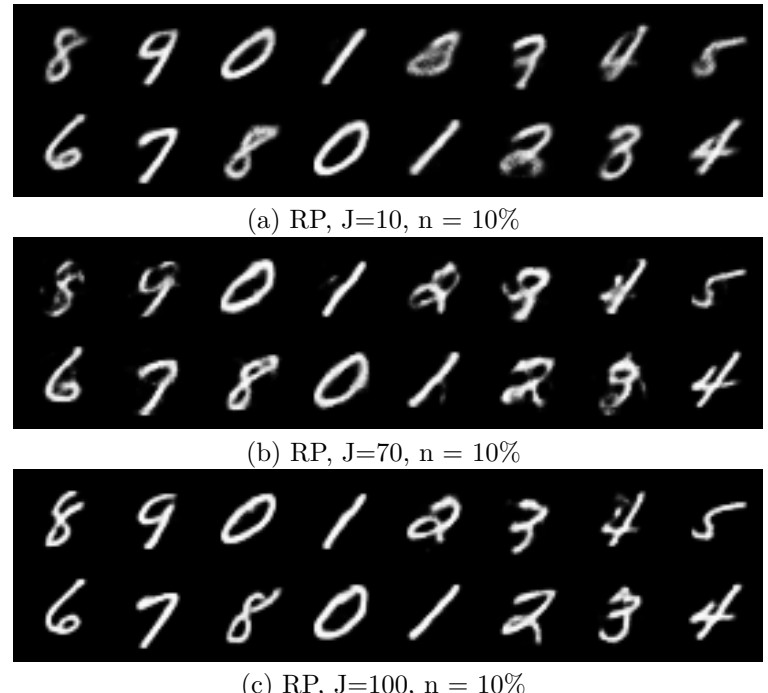

(a) RP, J=10, n = 10%

(b) RP, J=70, n = 10%

(c) RP, J=100, n = 10%

Figure 12: Stochastic Visualisation Of MNIST (RP Parameterisation)

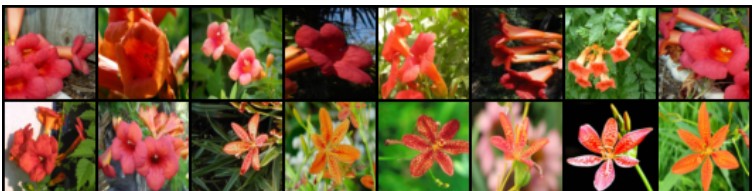

Figure 13: Ground-truth

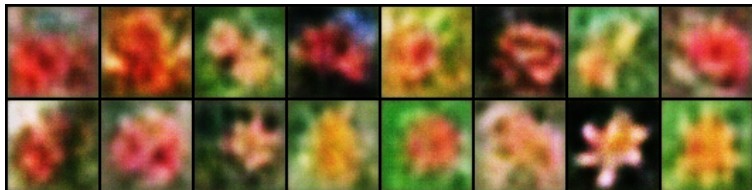

Figure 14: Full, J=100

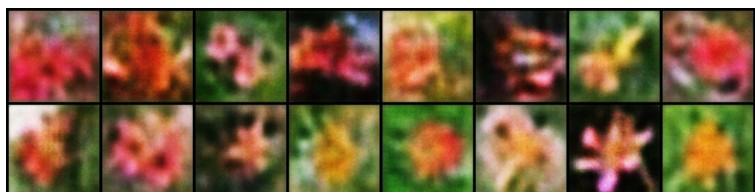

Figure 15: D, J=150

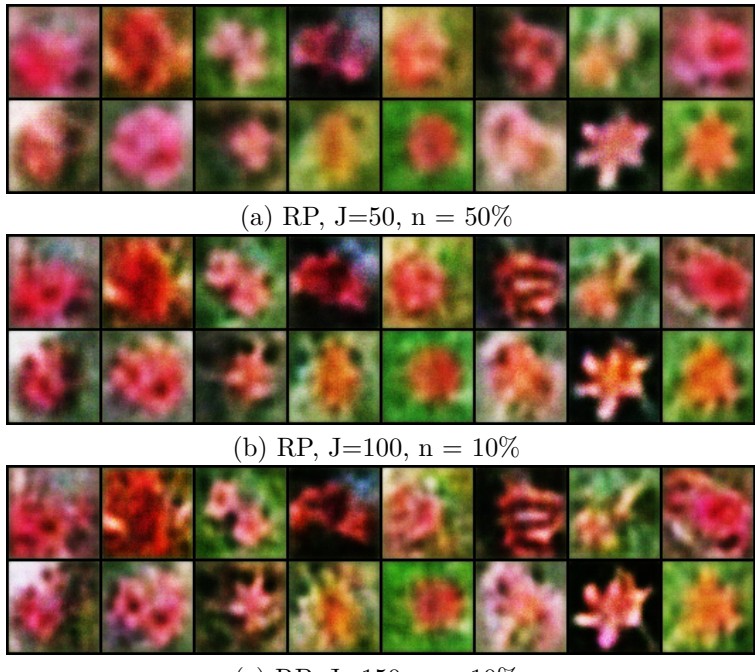

(a) RP, J=50, n = 50%

(b) RP, J=100, n = 10%

(c) RP, J=150, n = 10%

Figure 16: Stochastic Visualisation Of Flower-1O2 (RP Parameterisation)

