# OpenReview forum: "Random Projection Variational Auto-Encoders"
_TMLR — Rejected by TMLR_

### Review · Reviewer_rrRq · 2024-04-29

**Summary Of Contributions:**

This paper proposes a parameterization for covariance matrices of Gaussian variational posteriors, intended for use in VAEs. The parameterization uses random projections to reduce the number of parameters in the per-datum posterior approximation.

**Audience:**

No

**Broader Impact Concerns:**

No broader impact concerns.

**Claims And Evidence:**

No

**Requested Changes:**

I believe this paper needs significant revision before I could recommend it for acceptance to TMLR. This includes:

- Clearly defining all details of the proposed method, and in particular explaining how the per-datum projection matrices interact with the parameterized encoder, and why this is a correct way of defining an approximate posterior.
- Adding a discussion of the existing lower-rank covariance approximations for VAEs and how the proposed approach differs, and comparing the approach to these related previous techniques.
- Fixing the experimental claims and other statements throughout the paper to be clear and correct.

The paper would also be improved by considering modern VAE setups that attain reasonable generation quality (rather than using decade-old baselines). An alternative would be to explore more toy settings (e.g. simple 2D point datasets) where the differences between different methods can be more directly controlled.

Specific detailed comments:

- Page 1
  - "because they output a parameter estimate that is a mean, not a maximum": I don't think this statement makes sense. Bayesian methods output an estimated distribution, which you can summarize via any number of statistics including mean value or maximum a-posteriori likelihood.
  - "Numerical", "Variational": I don't think these should be capitalized.
  - "When using a Variational approximation, we find the best-fitting parametric model to a statistical distribution of interest (which is typically a likelihood)": This is confusingly written. I believe variational approximations typically find best-fitting approximations to posteriors, which are not likelihoods.
  - "overfitting": What sense of overfitting do you mean here? The variational bound should already accounts for overfitting due to the KL divergence term, so I don't see why you need to control overfitting separately.
- Page 2
  - "We propose a more flexible parameterisation that we perceive to be novel in this context." See my comments above; I believe there are many similar alternative parameterizations that are well-known in the community.
- Page 3
  - "Kingma et al. (2016). Kingma et al. (2016)" is repeated
- Page 4
  - "a Gaussian MLP": What do you mean by this?
- Page 6
  - "We do not use Full, which could be considering spurious correlations, or D, which assumes that all variables are independent within the covariance." What does this mean? It seems like you do use Full and D for the experiments (?)
- Figures on Page 7 and 8:
  - It's not clear what "n as % of J" is referring to. Is it supposed to be the labels on each point? If so, I don't think it should be on the Y axis.
  - What is the difference between figure 4 and figures 5, 6, 7?
  - What units are "runtime" denominated in?
- Page 9:
  - "sufficiently few parameters to avoid overfitting": What type of overfitting are you referring to? Why would increasing the expressivity of the covariance matrix lead to overfitting given that you already have a KL regularization term?
- Page 10:
  - What is "the ratio of spurious correlations"?
  - What does "Indicating that the latent space requires more than 150 × 1 parameters and less than (50 × (50 + 1)/2) to be expressed" mean?
  - What is "the trend observed in the MC mean Total loss presented in table 4"?

There are also many other minor grammatical issues throughout the paper; please carefully re-read your work for grammatical correctness.

**Strengths And Weaknesses:**

On the positive side, this paper considers an interesting problem (designing expressive alternative parameterizations for variational posteriors), and includes a number of experiments comparing their proposed approach to full or diagonal covariance matrices in terms of ELBO and runtime. Unfortunately, the paper has many significant weaknesses in its current form.

### Weakness 1. Method is not clearly explained and may not be well defined

I found it difficult to understand the details of how the authors apply their technique to the VAE setting. The posterior $q_\phi(z|x^{(i)})$ is claimed to be parameterized by $\phi$ and specified via a mapping $g_\phi(\epsilon^{(i,l)}, x^{(i)})$, which I believe the authors later refer to as an "encoder" and state to be a convolutional NN. However, this mapping doesn't seem to be explicitly defined. Later, the authors directly define the per-datum square-root-of-covariance matrix as $\Sigma^{(i)^{1/2}} = \Lambda_{J,n}^{(i)} Q_n^{(i)}$, and state that they "sample the projection for each datum and then consider this to be fixed for the duration of the optimisation". The authors also refer to the number of "per-datum parameters" of their method.

These two details seem inconsistent. If $q_\phi(z|x^{(i)})$ is a parameterized mapping, there should not be any per-datum parameters. Instead, the only parameters should be in $\phi$, and the per-datum covariance should be computed by running $g_\phi$ on $x^{(i)}$. Conversely, if the variational approximation is done on a per-datum level, there is no need for an encoder, and it doesn't make sense to compute a test loss (since parameters for the test data points are never learned).

The authors also seem to state that the $\epsilon^{(i,l)}$ values are held fixed for each datum (*"We consider the random projection in the same way as the reparameterised error input is considered in the approximation to the expected reconstruction error: we sample the projection for each datum and then consider this to be fixed for the duration of the optimisation."*). This seems like a misguided use of the variational approximation. In all VAE setups I'm aware of, optimization is done over samples from the approximate posterior, not a fixed set of predefined noise inputs.

### Weakness 2. Highly similar to existing covariance parameterizations (which are not discussed)

The motivation for the proposed approach is to provide flexible alternatives for covariance approximations between diagonal and full-rank predictions. However, the paper does not discuss similarities to existing structured covariance approximations used within the Bayesian neural network and probabilistic programming communities.

In particular, [Tomczak et al. (2020)](https://proceedings.neurips.cc/paper/2020/file/310cc7ca5a76a446f85c1a0d641ba96d-Paper.pdf) propose a variational inference technique very similar to the one proposed here, which uses a diagonal-plus-low-rank covariance factorization to improve flexibility without using a full-rank covariance. This diagonal-plus-low-rank structure is also built into the Edward probabilistic programming library as ["MultivariateNormalDiagPlusLowRank"](https://edwardlib.org/api/ed/models/MultivariateNormalDiagPlusLowRank), and is also available [in TensorFlow Probability](https://www.tensorflow.org/probability/api_docs/python/tfp/distributions/MultivariateNormalDiagPlusLowRankCovariance).

The difference between the proposed covariance here (using random projections) and the diagonal-plus-low-rank covariance techniques seems to be that the proposed method is more restricted: it doesn't include the diagonal terms and holds the low-rank subspace fixed instead of learning it. These seem like disadvantages of the proposed approach, and it doesn't seem like the proposed approach adds much, so I'm not sure it will be interesting to the TMLR audience.

I think at minimum this work should acknowledge and compare against these existing covariance factorizations, since they already accomplish the stated goal of finding a "'middle ground' between the theoretical robustness offered from a full rank covariance and the reduced computational complexity of the diagonal approximation."

(It's also worth noting that low-rank covariance approximations have been used elsewhere for covariance estimation, e.g. [Seeger (2010)](https://infoscience.epfl.ch/record/161304?v=pdf), [Hensman et al. (2013)](https://arxiv.org/abs/1309.6835), [Casale et al. (2018)](https://proceedings.neurips.cc/paper_files/paper/2018/file/1c336b8080f82bcc2cd2499b4c57261d-Paper.pdf))


### Weakness 3. Claims about experiments are confusing and not clearly unjustified

I could not understand many of the claims about the experiments in section 5, and I could not find clear evidence for them. For instance:

- *"It is more important for a covariance to consider correlations instead of assuming independent variances for its samples."*: What is important about it?
- *"the ratio of spurious correlations is highest when considering Full"*: I'm not sure what "ratio of spurious correlations" means, or why the experiments show that it is high. Spurious correlations between what? Spurious correlations usually refer to correlations between features and labels, but this is a generative task.
- *"Indicating that the latent space requires more than 150 × 1 parameters and less than (50 × (50 + 1)/2) to be expressed."*: This isn't a complete sentence and I'm also not sure what it means.
- *"both the KL divergence and expected reconstruction error follow the trend observed in the MC mean Total loss presented in table 4."*: It seems to me that the two error components have quite different trends from each other?

### Weakness 4. Many minor clarity and correctness issues throughout

I found many somewhat minor but distracting/confusing statements throughout the paper, and many grammatical and typographical errors. These are described in more detail in the "Requested Changes" section.

### Weakness 5. Experiments may not be relevant to modern usage of VAEs

The experiments all use small VAEs and small datasets based on the original VAE paper from more than ten years ago (Kingma & Welling, 2013). This means that the results are not representative of modern-day VAE usage, and the generated images in Figure 8 are quite bad for all methods.

The experimental claims are primarily based on the test loss, which I believe is an evidence lower bound on the generative log-likelihood of the test set. Since the models seem like quite bad generative models, I don't think the results here tell us much about whether this approximation technique is useful in practice.

---

> ### Author Response · Authors · 2024-05-24
> **Rebuttal for Reviewer 1**
>
> We appreciate Reviewer 1’s feedback and regret any misunderstanding of our method. Specifically, it appears there was confusion regarding the orthonormal matrix ($\Lambda$) in our approach. To clarify, we fix $\Lambda$ during the training of the algorithm, not $\epsilon$. Fixing $\Lambda$ allows us to train a low-dimensional full-rank covariance matrix, which we then project to the necessary dimensions, rather than training a larger full-rank matrix corresponding to the size of the latent space.
>
> With the help of reviewer 1’s extensive Requested Changes we are confident we can provide a clearer version of our work.
>
> Weakness 1:
>
>  It is apparent that our Method section requires re-formulation. We need to explicitly define our model alongside its parameterisations of the covariance and the vanilla VAE’s implementation and definitions (Kingma & Welling (2013)).
>
> We will do so with the Requested Changes in mind to ensure clarity and precision.
>
> Weakness 2:
>
> As we briefly discussed in the last paragraph of section 2 (Related work) we want to preserve the simplicity offered by Gaussian approximations. While we mentioned Normalizing Flows and their benefits and drawbacks compared to our method, we will extend the Related Work section to include the reviewer’s suggestions and additional citations.
>
> We find our method less restrictive, both in terms of expressing the distribution and computational efficiency, compared to existing works for several reasons. First, we infer both a non-isotropic diagonal term and the correlations within the low-dimensional space. Additionally, based on feedback from Reviewers 1 and 2, we have enhanced our method by incorporating a second non-isotropic diagonal term with the latent space dimensions. This results in a more accurate mathematical implementation rather than an approximation. These three terms—the initial diagonal, the correlation, and the larger dimensional diagonal—are all inferred by their gradients without constraints using the Adam optimizer. A key feature of $\Lambda$ is its orthonormality, which facilitates the efficient projection of information into higher-dimensional spaces (Mervin E muller (1959)), thereby better capturing variances correlations within the lower-dimensional space, especially by preserving the structure of the covariance matrix through its (orthogonal) projections (in contrast with Hensman et al. (2013)). Computationally, our approach eliminates the need to compute substantial additional gradients (as in Normalizing Flows) or eigenvalues (as in PCA, Seeger (2010)) to train our algorithm.
>
> These aspects will be detailed in the revised manuscript with appropriate citations.
>
> Weakness 3 – Weakness 4:
>
> We need to express our findings more carefully within the discussions and through the whole document. Once again, we thank the reviewer for their extensive Requested Changes section of the review.
>
> Weakness 5:
>
> The reviewer makes a valid point regarding the need to test our method against state-of-the-art approaches to yield more compelling and visually pleasing results.
>
> We will explicitly state, not just in the Abstract but throughout the paper, our intent to provide an ablation study featuring a ‘stripped down’ version of the VAE without any enhancements to offer a stricter comparison of the different parameterizations we discuss.
>
> We chose MNIST and CIFAR10 for their ease of comparison with other current research. While our results may not significantly deviate from other approximations in terms of metrics, the benefits of our approach are evident from the tables and particularly from the trends in the figures (1-3 and 5-7). Particularly interesting is that we included Flower102 to highlight scenarios where our parameterisation and others would struggle, revealing interesting metrics (Table 4-5) and trends (Figure 3).
>
> As mentioned in section 2 (Related Work) our parameterisation could be applied to any work that utilises diagonal approximations or full-rank covariances, including improved VAEs and BNN. We are eager to explore these applications in future research.
>
> Overall:
>
> With reviewer 1’s comprehensive and helpful feedback, we are well-equipped to revise the paper, incorporating all recommendations to enhance clarity and impact. Thank you once again for your valuable insights.

---

> > ### Comment · Reviewer_rrRq · 2024-06-08
> > **Response to rebuttal**
> >
> > Thank you for your response. It seems that you are planning to make substantial changes to the work beyond the currently submitted version. I agree that explicitly defining the model in the method section will improve the clarity of the work, and will hopefully address some of the concerns in my review. I still believe that it is important to directly compare to other existing VAE parameterizations as well, given how similar the proposed work is to existing alternative factorizations.
> >
> > Since many changes are still required for me to be satisfied that this work meets the TMLR acceptance criteria, I maintain my original review.

---

### Review · Reviewer_o21F · 2024-05-10

**Summary Of Contributions:**

This manuscript looks at learning a structured posterior in a variational auto encoder by using a random projection on the variance from a small compressed latent space to the full latent space.  The general idea is that the compressed nature of the posterior allows more tractable inference that can lead to better solutions.

**Audience:**

Yes

**Claims And Evidence:**

No

**Requested Changes:**

The authors need to fully justify their mathematical derivation, or the authors need to adopt a low-rank + diagonal representation to give a full-rank posterior covariance matrix.

**Strengths And Weaknesses:**

Strengths:

I found this manuscript clearly written and easy to read.  There are numerous experiments exploring the predictive performance.

Weaknesses:

I was surprised by the chosen structure of the posterior, where the variance term as defined in (4) is not full rank.  As such, I do not see how the  KL divergence term defined in (6) is correct.  In particular, the full posterior variance is $ \Lambda_{J,n}^{(i)} Q_n^{(i)}(Q_n^{(i)})^T  (\Lambda_{J,n}^{(i)})^T$, and then the $\log |\Sigma^{(i)}|=\log |\Lambda_{J,n}^{(i)} Q_n^{(i)}(Q_n^{(i)})^T  (\Lambda_{J,n}^{(i)})^T|$, as the authors state in (24)  As the matrix is not full-rank, then the determinant is 0, and $\log 0 \rightarrow -\infty$, not the progression from (24) to (25).  Thus, as stated above, I do not see how the approximation the authors propose is appropriate or correct, and at a minimum needs to be significantly more justified.  I would have expected alternatively that the used form for the covariance matrix would be diagonal + low-rank, e.g., $\Lambda_{J,n}^{(i)} Q_n^{(i)}(Q_n^{(i)})^T  (\Lambda_{J,n}^{(i)})^T+\text{diag}(\sigma^{(i)}\odot\sigma^{(i)})$, which is full rank.  The KL could still be calculated efficiently on such a form through linear algebra tricks.

As a side note, it is possible to simplify the trace through its cyclic property, so I do not agree with the claim above (26).

The authors note that they used the same convolutional layers as in Kingma and Welling (2013), but Kingma and Welling do not use convolutional layers.  Please correct.

---

> ### Author Response · Authors · 2024-05-24
> **Rebuttal for Reviewer 2**
>
> We are grateful for Reviewer 2’s feedback, which identified a significant issue in our mathematical derivations. Specifically, after projecting the lower full-rank covariance matrix to the latent space, it is no longer considered a full-rank matrix, resulting in an incorrect calculation of the KL-Divergence.
>
> Fortunately, we had also considered and implemented an alternative version of the covariance + diagonal approximation, denoted as RP+D: $\Lambda_{J,n}^{(i)} Q_n^{(i)}(Q_n^{(i)})^T (\Lambda_{J,n}^{(i)})^T + \text{diag}(\sigma^{(i)} \odot \sigma^{(i)})$. From an engineering perspective, we initially regarded RP+D as an overparameterization and chose not to include it, as the metric results differed, but the overall discussion and hypothesis remained unchanged.
>
> We have already executed the same experiments using this version of the KL-Divergence computation and are prepared to substitute the RP parameterization with RP+D.
>
>
>
> We would also like to point out that we find it intriguing that despite the mathematical issue discussed above, the determinant of the ‘approximated’ covariance matrix was not zero in any of the practical tests we performed. We plan to investigate this phenomenon further in future work.
>
> Additionally, the reviewer correctly pointed out that we use convolutional layers instead of linear layers, and we will make the necessary corrections in our manuscript.

---

> > ### Comment · Reviewer_o21F · 2024-06-10
> > **Rebuttal Acknowledgement**
> >
> > Thank you for your feedback.  I encourage the authors to consider that revised approach, which would be a major revision throughout the manuscript.  Without such edits, I maintain my original review.

---

### Review · Reviewer_vJf4 · 2024-05-21

**Summary Of Contributions:**

The paper tackles the problem of covariance matrix parametrization for the variational posterior density of "Gaussian" VAEs. The driving idea is that instead of optimizing all the parameters of a full-rank covariance matrix or assuming a diagonal structure approximation, one can instead introduce a linear projection on to a sub-space of lower dimensionality. The paper explores this direction on well-known datasets and brings sets certain conclusions from the exploratory results.

**Audience:**

No

**Claims And Evidence:**

No

**Requested Changes:**

In general, I think that the work is in an extremely early stage, with some exploratory results of the idea of random projections. Related work, comparison, empirical results, and discussion are limited and under the quality threshold for a journal like TMLR. The manuscript requires a lot of improvement to its current state to be close to acceptance (in my opinion). I also want to remark that the random projection idea seems to be just an extremely well-known linear algebra operation applied to one classic ML model that does not really struggle with the dimensionality of the parameters in the latent posterior as it is claimed. I hope that the authors understand that this means that the novelty is very limited right now in this regard.

**Strengths And Weaknesses:**

- The paper motivates a lot on Bayesian NNs, and justifies the introduction of the lower dimensionality projection on the computational cost and the number of parameters to be optimized. However, this view is odd here, as the number of parameters does not grow as fast here or is not as challenging as in BNNs (where the posterior is on the parameter space that is difficult to keep it reduced when we treat with NNs). Notice that the covariance matrix that is being modelled here is on the posterior over the latent variables (which we can control easily and typically does not grow that much as the number of NN's parameters).

- The works skips dozens of well-known works and approximations to the covariance matrix via linear algebra or other methods. How is that possible that KFAC or block-wise methods are not considered, referenced, or used for comparison? Having a lack of reference to this sort of method, I find difficult to trust the related work included and I do think that (very likely) previous methods have explored this sort of simple projections before in several contexts of ML.

- Experiments are extremely limited, with no comparison and on very simple datasets. The losses do not indicate a decent performance or significant improvement. The results just look as an empirical observation of the well-known effect of latent space dimensionality in VAEs.. The discussion does not bring any significant insight for the reader of TMLR in the current state.

---

> ### Author Response · Authors · 2024-05-24
> **Rebuttal for Reviewer 3**
>
> We thank Reviewer 3 for their feedback, which we consider valuable for any future work on this research. Although we mention the potential application of the RP parameterisation method for BNNs, this is not the scope of the current paper. Our current focus is on demonstrating that the RP method can serve as an alternative parameterisation, balancing computational expense, overfitting, and accuracy between full-rank and diagonal covariances.
>
> With this in mind, we respond to the feedback regardless.
>
> Preferring the RP parameterisation over the diagonal approximation, which is commonly used for BNNs due to the volume of parameters in neural networks, should provide an acceptable approximation to the full-rank covariance. Our proposed approach improves accuracy (as shown in Tables 2-4 and Figures 1-3) while slight compromising on time complexity (Figures 4-7).
>
> We are rather intrigued to find out how much so in a future work.
>
> To the best of our knowledge, Random Projection has not been used to describe a learnable lower dimensional full-rank covariance with Variational Inference in mind. The KFAC method referenced by the reviewer is well-regarded within the optimisation community. However, there are key differences compared to our proposed method. For instance, orthogonal projections in RP can capture complex dependencies within the data more effectively than KFAC’s block-diagonal structure. KFAC offers balance by providing a structured approximation that retains critical information for optimisation. RP might be more suitable for scenarios where preserving the full-rank structure in a transformed space is crucial, whereas KFAC is beneficial in standard deep learning training processes where efficiently approximating the FIM can lead to better convergence properties. However, these considerations are outside the scope of the current paper.
>
> If we were to apply RP during optimisation of BNNs we would gladly consider the reviewer's proposition to compare it with KFAC.
>
> Regarding the simple datasets, assuming the reviewer refers to MNIST and CIFAR10, we chose them because they are widely used by researchers, facilitating comparison between studies. Our goal is not to achieve the best algorithmic performance but to conduct an ablation study on the impact of different parameterizations on expressing a distribution, as mentioned in the Abstract.

---

### Decision · Action_Editor_1b1h · 2024-06-10

**Recommendation:** Reject

**Comment:**

This paper proposes a method to learn low rank structure in the covariance matrix of latent variables in variational autoencoders. While the topic is of interest to TMLR's community, the paper (1) contains errors, and (2) does not properly discuss related work. In particular, well-known related work has already proposed a similar but better motivated scheme, so even if the authors were to properly cite this work, I do not believe the paper would be of interest to TMLR's audience. I thus believe the paper should be rejected with no option to be resubmitted to TMLR.

**Audience:**

While the topic of the paper is of interest to TMLR's audience, the findings in the paper are not.

**Claims And Evidence:**

This paper does not provide evidence to back up its claims: it contains mathematical errors, and ignores extremely relevant related work.